# Cow Milk Extracellular Vesicle Effects on an In Vitro Model of Intestinal Inflammation

**DOI:** 10.3390/biomedicines10030570

**Published:** 2022-02-28

**Authors:** Samanta Mecocci, Alessio Ottaviani, Elisabetta Razzuoli, Paola Fiorani, Daniele Pietrucci, Chiara Grazia De Ciucis, Silvia Dei Giudici, Giulia Franzoni, Giovanni Chillemi, Katia Cappelli

**Affiliations:** 1Department of Veterinary Medicine, University of Perugia, 06123 Perugia, Italy; samanta.mecocci@studenti.unipg.it; 2Sports Horse Research Center (CRCS), University of Perugia, 06123 Perugia, Italy; 3Department of Biology, University of Rome Tor Vergata, 00133 Rome, Italy; alessio.ottaviani@uniroma2.it; 4National Reference Center of Veterinary and Comparative Oncology (CEROVEC), Istituto Zooprofilattico Sperimentale del Piemonte, Liguria e Valle d’Aosta, Piazza Borgo Pila 39-24, 16129 Genova, Italy; chiaragrazia.deciucis@izsto.it; 5Institute of Translational Pharmacology, National Research Council, CNR, 00133 Rome, Italy; paola.fiorani@uniroma2.it; 6Institute of Biomembranes, Bioenergetics and Molecular Biotechnologies, IBIOM, National Research Council, CNR, 70126 Bari, Italy; daniele.pietrucci@unitus.it; 7Department for Innovation in Biological, Agro-Food and Forest Systems (DIBAF), University of Tuscia, 01100 Viterbo, Italy; 8Department of Animal Health, Istituto Zooprofilattico Sperimentale della Sardegna, 07100 Sassari, Italy; silvia.deigiudici@izs-sardegna.it (S.D.G.); giulia.franzoni@izs-sardegna.it (G.F.)

**Keywords:** EVs, bovine milk, IBD, RT-qPCR, Caco-2, THP-1, co-cultured, model of intestinal inflammation, anti-inflammatory, immunomodulating

## Abstract

Extracellular vesicles (EVs) are lipid bilayer nano-dimensional spherical structures and act mainly as signaling mediators between cells, in particular modulating immunity and inflammation. Milk-derived EVs (mEVs) can have immunomodulatory and anti-inflammatory effects, and milk is one of the most promising food sources of EVs. In this context, this study aimed to evaluate bovine mEVs anti-inflammatory and immunomodulating effects on an in vitro co-culture (Caco-2 and THP-1) model of intestinal inflammation through gene expression evaluation with RT-qPCR and cytokine release through ELISA. After establishing a pro-inflammatory environment due to IFN-γ and LPS stimuli, *CXCL8, IL1B, TNFA, IL12A, IL23A, TGFB1, NOS2,* and *MMP9* were significantly up-regulated in inflamed Caco-2 compared to the basal co-culture. Moreover, IL-17, IL-1β, IL-6, TNF-α release was increased in supernatants of THP-1. The mEV administration partially restored initial conditions with an effective anti-inflammatory activity. Indeed, a decrease in gene expression and protein production of most of the tested cytokines was detected, together with a significant gene expression decrease in *MMP9* and the up-regulation of *MUC2* and *TJP1*. These results showed a fundamental capability of mEVs to modulate inflammation and their potential beneficial effect on the intestinal mucosa.

## 1. Introduction

Extracellular vesicles (EVs), nano-dimensional spherical structures enclosed by a lipid bilayer membrane, include many types of vesicles that differ for the biogenesis process, dimension, tissue of origin, and function. Indeed, three main subtypes can be identified: exosomes, the smallest one (30–100 nm), generated from the fusion of multi-vesicular bodies membrane (cellular organelles composed of intraluminal vesicles through their membrane inward budding) with the cellular membrane; microvesicles, also known as ectosomes or shedding vesicles, released through the exocytosis process and ranging from 100 to 1000 nm; and apoptotic bodies greater than 1000 nm [1,2,3]. Furthermore, vesicles associated with a specific tissue or a particular function were discerned, such as prostasomes, microparticles, tolerosomes, and oncosomes [4,5,6,7,8,9].

The confusion in nomenclature generated from the increasing number of works in the EV field, associated with the difficulties in distinguishing these different types of vesicles through the current techniques, made the International Society of Extracellular Vesicles solicit the use of the generic EV, as stated in the developed guidelines [10].

EVs function as signaling mediators between cells with autocrine, paracrine, juxtacrine, and endocrine activity, thus released in the extracellular environment by all cell types and found in any biological fluids [11]. The regulation in receiving cells is mediated by many molecules contained within EVs such as proteins, antigens, lipids, metabolites, RNAs, and recently found DNA fragments [11,12,13]. The complex cargo induces a wide range of functional modulation in receiving cells, depending on the type of recipient cells and the stimuli that these cells receive [14,15]. In particular, a role in immunomodulation to both immune and non-immune cells related to antigen-specific and non-specific activation has been observed [11,16,17,18]. EVs can also modulate inflammation and have been found implicated in the pathogenesis of chronic inflammatory diseases such as diabetes, arthritis, and inflammatory bowel disease (IBD) [19,20,21]. IBD includes a group of autoimmune diseases that affect the gastrointestinal tract, such as ulcerative colitis (UC) and Crohn’s disease (CD). These pathologies are characterized by prolonged inflammation coupled with immune dysregulation. IBD has become a real health problem in the last decades because of a significant increase, especially in industrialized and neo-industrialized countries where it constitutes one of the most prevalent gastrointestinal diseases [22]. The reasons for this spread lie in pathogenetic mechanisms challenging to fully understand since many actors intervene in this dysregulation and the lack of effective therapies [23]. Genetic susceptibility of the host seems to be an important factor, even though only a small portion of individuals that carry IBD-associated risk loci develops IBD [24]. Therefore, additional factors are implicated, such as alterations of the interactions between the gut microbiota and the mucosal immune system. In fact, an imbalance in the immune response of both innate and adaptive immunity consequent to an altered interaction with the microbiome can cause IBD [25,26]. In this exchange, the mucosal barrier function is crucial to maintain the homeostatic balance [27] since its dysfunction and dysbiosis have been correlated with IBD [28]. In animal models, it has been demonstrated that intestinal microbiota plays both pro-inflammatory and anti-inflammatory roles on the intestinal mucosa, clearly promoting the development of IBD; in humans, a direct cause-effect is more difficult to assess, but many specific microbes have been associated with these pathological conditions, especially in Crohn’s disease patients [24]. Moreover, environmental factors such as food, smoke, stress, and medical conditions can influence the IBD onset [24].

In this scenario, it is entirely legitimate to hypothesize a role for EVs in the pathogenesis of IBD within the intestinal microenvironment, given their intrinsic function in cell communication and the capability of being produced by all cell types. Indeed, in recent years, the role of EVs in IBD has been debated and has garnered interest among researchers, actually encountering the ability of EVs to regulate communications between the already cited actors through proteins, RNAs, and lipids carried into the cargo [29].

At the same time, the EV features can be exploited as a tool for modulating the intestinal elements, repairing damage, and restoring intestinal mucosal barrier functions. Indeed, their non-toxicity, biocompatibility, organotropism, and targeting ability have brought to the development of a specific research branch in the EV field focused on their use as nanocarriers of biomarkers for disease diagnostics or as a delivery system for targeted therapeutic approaches [30,31]. Recently, many manipulation technologies have been implemented for engineering EVs also for IBD treatment, enhancing their therapeutic capability or using them as nanocarriers for drug delivery [19]. Many studies highlighted the ability of EVs to regulate immune cells and cytokines within the inflammatory microenvironment, dampening inflammation and restoring the intestinal barrier integrity and gut microbiome composition and diversity [19]. Evidence of these EVs improving effects on IBD emerged from in vivo studies on a dextran sodium sulfate-induced IBD animal model, where MSC-derived EV administration decreased the disease severity, reducing inflammation and improving epithelial functions [32,33]. Some studies on EVs derived from food showed ameliorating properties on intestinal inflammation for those isolated from edible plants such as *Curcuma longa*, grapes, and broccoli [34]. One of the most promising food sources of EVs in terms of quantity is indubitably milk, and milk-derived EVs (mEVs) have been investigated in humans and in many mammals such as cows, buffalo, pigs, camels, and sheep [35,36,37,38,39]. In some cases, a direct action on the intestine was shown. In fact, mEVs were able to enhance intestinal cell growth, reduce cell death related to lipopolysaccharides (LPS), increase mucin production, and promote intestinal microbiota [34]. These capabilities can be really feasible in vivo since mEVs can successfully cross the harsh condition of the gastrointestinal tract, reach and be absorbed by the intestinal cells [40,41,42]. Moreover, milk is essential for infant mammals: it provides nutrition and helps the development of the neonatal gut immune system through maternal immunoglobulin transmission [43,44]. Recently, EVs’ described immunomodulation role has also been confirmed in mEVs, accompanied by anti-inflammatory functions [45,46], suggesting a sort of adjuvant activity in the newborn immune system shaping. In fact, mEVs are rich in bioactive molecules such as proteins, lipids, RNAs, and metabolites, which also emerged from our previous studies, known to influence inflammation and the immune system [47,48,49]. These effects have been demonstrated in human peripheral blood mononuclear cells, T regulatory cells, and macrophages [50,51], but a more comprehensive approach, where the impact of mEVs on gut inflammation is evaluated, taking into account the interaction of intestinal and immune cells, is missing.

In this context, this study aimed to evaluate the anti-inflammatory and immunomodulating effects of mEVs derived from bovine milk on an in vitro co-culture model of intestinal inflammation [52].

## 2. Materials and Methods

The experimental design underlying this study and the main methodical passages applied is schematically explained in Figure 1.

### 2.1. Milk Collection

Cow raw milk samples were collected from mass milk to cope with the individual variability. The raw milk was sampled from the Didactic Zootechnical Farm surveilled by the Veterinary Medicine Department, University of Perugia. Holstein Friesian cattle breed was chosen for milk sampling, and the majority of the animals were in mid-lactation. Rearing conditions are referable to standard intensive farms with unifeed and mechanical milking. Milk was immediately processed, avoiding cryo-preservation to minimize artifacts.

### 2.2. Extracelluar Vescicles Isolation

For mEV isolation, we exactly followed the protocol of our previous study (a schematic illustration is shown in Figure 1A), where vesicle isolation was verified through morphological characterization with transmission electron microscopy (TEM), Western blotting, and nanoparticle tracking assay (NTA) [48]. In brief, two consecutive 3000× *g* centrifugations for 10 min at room temperature (RT) (Eppendorf^®^ Centrifuge 5810R with an F34-6-38 rotor) were applied to 300 mL of raw milk to eliminate fat globules in the upper layer and cells and cell debris in the pellet; then 0.25 M ethylenediaminetetraacetic acid (EDTA) (pH 7.4) was added to the supernatant in a 1:1 ratio, incubated for 15 min on ice, and centrifuged at 10,000× *g* for 1 h at 4 °C. Centrifugation at 35,000× *g* for 1 h at 4 °C was carried out for the supernatant in a Beckman Coulter Optima L-100 XP with a 45 Ti rotor, and final ultracentrifugation at 200,000× *g* for 90 min at 4 °C was applied collecting mEVs in the pellet. Pellets were resuspended in 1× phosphate-buffered saline (PBS) (Euroclone, Pero, Italy) sterile filtered (0.22 μm pore size) and conserved at −80 until use. The suspensions were used within a week from the isolation.

### 2.3. Size Distribution of EVs and Assessment of Concentration to Be Administered

Isolated mEVs were then measured in terms of size distribution and concentration through the Malvern Panalytical NanoSight NS300 NTA system (Malvern, Worcestershire, U.K.) (Figure 1B and Figure 2). Before measuring, the mEV suspension derived from one pellet (approximately 15 mL of raw milk) was further diluted and filtered (0.22 μm pore size) in 1× PBS (Sigma, St. Louis, MO, USA) to reach the most suitable concentration for the NTA system, and five measurements were performed. Results are reported as mean +/−1 standard error of the mean. To set up the concentration of mEVs to be administered to cell cultures, cell viability was tested through the 3-(4,5-dimethylthiazol-2-yl)-5-(3-carboxymethoxyphenyl)-2-(4-sulfophenyl)-2H-tetrazolium, inner salt (MTS) assay, and the cytotoxicity and cell damage in the co-culture was evaluated through the quantification of lactate dehydrogenase (LDH) release.

### 2.4. Cell Cultures

Caco-2 cells were maintained in RPMI-1640 medium supplemented with 10% heat-inactivated FBS, 2 mM L-glutamine, 0.1 mg/mL streptomycin, and 100 U/mL penicillin (Euroclone, Pero, Italy) hereafter indicated as complete media (CM).

The THP-1 cells were kindly donated by Professor Maria Rosa Ciriolo (University of Rome Tor Vergata). Cells were cultured in RPMI-1640 CM. Both cell lines were grown in an incubator at 37 °C under 5% CO_2_.

For MTS and LDH assay, Caco-2 cells were cultured for 21 days to allow differentiation, while THP-1 cells were treated the day before the experiment with 10 ng/mL phorbol 12-myristate 13-acetate (PMA) (MERCK, Darmstadt, Germany). For co-cultures experiment, the protocol of Kämpfer and collaborator with slight modification was followed [52]. In brief, 0.2 × 10^6^ Caco-2 cells were seeded on transwell inserts (1 μm pore size; Greiner Bio-one Kremsmünster, Austria) of a 12-well plate and maintained for 21 days in RPMI-1640 CM to allow cells differentiation. The day before the assay, 0.2 × 10^6^ THP-1 cells were seeded in a 12 wells plate in RMPI-1640 CM supplemented with 10 ng/mL PMA to be differentiated into macrophages, while Caco-2 cells were treated with 10 ng/mL IFN-γ (Peprotech, London, UK) in the baso-lateral (BL) compartment of the plate for 24 h to impair the barrier integrity. The insertion of the transwell containing Caco-2 cells in the multiwell with differentiated THP-1 cells, without any other treatment, was considered as “C0” and used to set up the baseline co-culture condition (Figure 1C).

To increase the Caco-2 barrier disruption, THP-1 cells were pre-stimulated with 10 ng/mL IFN-γ and 10 ng/mL LPS (MERCK, Darmstadt, Germany) 4 h before the co-culture was initiated with IFN-γ-primed Caco-2 cell layers. Before starting the co-culture, THP-1 cells were washed with 1× PBS to remove IFN-γ and LPS and replaced with fresh RPMI-1640 CM. After THP-1 stimulation, the transwell containing Caco-2 cells were moved into the 12-well plates with THP-1 cells in the BL compartment and co-cultured for 24 h in a 37 °C incubator with 5% CO_2_ to start the inflammation. Hereinafter, this set-up with 24 h-differentiated THP-1 cells, 4 h pre-exposed to LPS and IFN-γ, in co-culture with 24 h IFN-γ-primed Caco-2 cells was considered as inflamed co-culture and named “C1” (Figure 1C).

After being co-cultured in previously described inflammatory condition for 24 h, cells were treated with two different concentrations of mEVs suspension (Figure 1C): 108 or 1010 were added to Caco-2 cells in the apical (AP) compartment for 24 h, indicated as treatment “C2a”; 108 and 1010 were added to both AP and BL compartments for 24 h, treatment “C2b”.

The experiment was repeated four times, with three technical replicates each experiment, using mEV isolates derived from different milk samples.

#### 2.4.1. MTS Assay

MTS assay was performed according to the manufacturer’s protocol (CellTiter 96^®^ AQueous One Solution Cell Proliferation Assay, Promega, Madison, WI, USA). Briefly, 0.1 × 10^6^ Caco-2 or THP-1 cells were plated on a 96 wells plate in RPMI-1640 CM. Caco-2 cells were cultured for 21 days in RPMI-1640 CM to allow cells differentiation. The day before the assay, THP-1 cells were plated and treated with 10 ng/mL of PMA. Cells were then incubated for 24 h at 37 °C, with 5% CO_2_. On the day of the assay, THP-1 was washed with PBS 1X to remove PMA. Both cell lines were treated with different amounts of mEVs starting from 1 × 10^4^ until 1 × 10^11^. Cells were then incubated again at 37 °C, 5% CO_2_ for 48 h. After the incubation, the media was removed and replaced with fresh RPMI-1640 CM supplemented with MTS reagent diluted 1:5 in fresh CM. The plates were then incubated again at 37 °C, 5% CO_2_ for 2 h, and the absorbance was measured at 490 nm using a microplate reader (TECAN). This test has been performed only for monocultures cells, while an LDH assay was performed in the case of co-cultures.

#### 2.4.2. LDH Assay

LDH releasing from co-culture cells was evaluated using Invitrogen™CyQUANT™ LDH Cytotoxicity Assay kit (Thermo Fisher Scientific, Waltham, MA, USA). THP-1 cells and/or Caco-2 cells were treated for 24 h with 1 × 10^10^ or 1 × 10^8^ mEVs and 50 μL of cell-free supernatant were added to 96-well plates with 50 μL of the reaction mixture and incubated for 30 min at RT protected from light. As a control, non-treated monocultured cells were used for the baseline, while C0 and C1 referred to conditions as reported in Figure 1C. After incubation, 50 μL of STOP solution were added to the plate and measured the absorbance at 490 and 680 nm using a microplate reader (TECAN) to determine LDH activity. The obtained data were normalized to maximum LDH release according to manufactured instruction.

### 2.5. RNA Extraction and RT-qPCR of Caco-2 Cells

Total RNA was extracted from differentiated Caco-2 cells at previously defined experimental conditions (Figure 1C,D), from transwell insert, using RNeasy Mini Kit (Qiagen s.r.l., Milan, Italy) through the Qiacube System (Qiagen s.r.l., Milan, Italy) in accordance with the manufacturer’s instructions. RNA extraction was assessed using a Nanodrop 2000 Spectrophotometer and Qubit 3.0 Fluorometer (Thermo Fisher Scientific, Waltham, MA, USA).

The same amount of RNA for each sample (500 ng) was reverse-transcribed into cDNA, using the SuperScript^®^ VILO IV TM Master Mix (Thermo Fisher Scientific, Waltham, MA, USA), following the manufacturer’s guidelines. Amplification was performed on CFX96™ Real-Time System (Bio-Rad, Milan, Italy) following a protocol developed in previous studies [53,54]. Primers of target and reference genes were designed in accordance with the sequences available on the Primer-BLAST online design platform (https://www.ncbi.nlm.nih.gov/tools/primerblast/, accessed on 31 July 2021), and primer pairs were placed in different exons or at exon-exon junctions in order to avoid biases due to genomic DNA amplification. Specific primer pairs for the reference genome were verified in silico using In-Silico PCR software (https://genome.ucsc.edu/cgi-bin/hgPcr, accessed on 31 December 2021) to confirm their specificity for targeting. Primer sequences of target genes, C-X-C motif chemokine ligand 8 (*CXCL8*), interleukin 1 beta (*IL1B*), *IL6*, *IL12A*, *IL23A*, tumor necrosis factor alpha (*TNFA*), matrix metallopeptidase 9 (*MMP9*), mucin 2 (*MUC2*), nitric oxide synthase 2 (*NOS2*), transforming growth factor beta 1 (*TGFB1*), trefoil factor 3 (*TFF3*) and tight junction protein 1 (*TJP1*), and reference genes, ribosomal protein lateral stalk subunit P0 (*RPLP0*), ribosomal protein L37a (*RPL37A*) and ribosomal protein S14 (*RPS14*), are reported in Table 1.

A preliminary RT-qPCR reaction efficiency test was performed for each primer pair. Moreover, in order to verify the amplification of specific products or primer dimer artifacts, the melt curves were examined, and a single well-defined peak was observed in the negative first derivative plot. Finally, reference genes were tested for their stability under different conditions.

### 2.6. Cytokine Levels Determination

Co-culture supernatants from AP and BL sides were removed at a previously defined time point (Figure 1C,D). Supernatants were cleared from debris by centrifugation (2000× *g* for 3 min) and stored at −80 °C until analysis. Levels of IL-1α, IL-1β, IL-6, IL-8, IL-10, IL-12p70, IL-12p40, IL17, TNF-α were determined using MILLIPLEX^®^ Premixed Human Cytokine Panel A (Merck Millipore, Darmstadt, Germany) and a Bioplex MAGPIX Multiplex Reader (Bio-Rad, Hercules, CA, USA), according to the manufacturers’ instructions, as previously described [47]. The obtained data were generated from the four independent experiments with 2 technical replicates.

### 2.7. Statistical Analysis

#### 2.7.1. MTS and LDH Assays

Results were generated from 4 independent experiments with 3 technical replicates each. The statistical and graphical analyses were performed using GraphPad Prism 6 (GraphPad Software Inc., La Jolla, CA, USA), and variations between results were expressed as standard deviation (S.D.). After checking the normality distribution through the Kolmogorov–Smirnov test, the data were statistically analyzed by one-way ANOVA.

#### 2.7.2. RT-qPCR

A normalization step was performed according to the expression levels of the three reference genes (*RPLP0*, *RPL37A,* and *RPS14*) after assessing their stability under different experimental conditions, using the Norm algorithm included in Bio-Rad CFX Maestro software (ver. 4.1 Bio-Rad, Hercules, CA, USA). Relative normalized expression was assessed using the 2^−∆∆Ct^ method [55], comparing the inflamed co-culture (C1) with basal co-culture C0 and inflamed co-culture with mEV administration C2a and C2b (Figure 1C,D). The data were analyzed using GraphPad Prism 5.04 (GraphPad Software Inc., La Jolla, CA, USA). Gene expression data were submitted to a Kolmogorov–Smirnov test to check Gaussian distributions. Significant differences were checked by Kruskal–Wallis test and applying the post-doc Dunn’s multiple comparison test. The significance threshold was set at *p* < 0.05.

#### 2.7.3. Cytokine Levels Determination

Results were generated from 4 independent experiments with 2 technical replicates each on supernatants of C0, C1, C2a, and C2b conditions (Figure 1C,D). First, data normality was checked using the Kolmogorov–Smirnov test. Then data were graphically and statistically analyzed with GraphPad Prism 8.01 (GraphPad Software Inc., La Jolla, CA, USA). Results were shown as box-plot and whiskers (min-max). The parametric one-way ANOVA followed by Dunnett’s multiple comparison test or the non-parametric Kruskal–Wallis test followed by Dunn’s multiple comparison test was used for statistical analyses, comparing all the other conditions vs C1. A *p*-value < 0.05 was considered statistically significant.

## 3. Results

### 3.1. mEV Size Distribution and Concentration

From NTA assay, the cow mEV mean (±1 standard error) diameter was 142.7 ± 2.9 nm, with a size distribution characterized by a unique peak and a narrow range (D10 = 92.6 ± 1.7 nm and D90 = 215.2 ± 4.1 nm), revealing a high homogeneity (Figure 2). Nanoparticle densities were reported as mean concentration (particles/mL) (±SD) of five measurements after one pellet (originated from about 15 mL of raw milk) resuspension in 400 μL of 1× PBS that resulted in 1.22 × 10^12^ (±3.63 × 10^10^).

### 3.2. mEV Effects on Cell Viability

To evaluate mEVs effect on cell viability, a cell viability assay was performed on 21 days differentiated Caco-2 cells and THP-1 cells differentiated with PMA. Both cell lines were treated with different mEV concentrations ranging from 10^4^ to 10^11^. The 0.1 × 10^6^ Caco-2 or THP-1 cells were seeded in a 96-well plate as described in material and method, and, subsequently, mEV administration, cells were incubated for 48 h in a 37 °C incubator under 5% CO_2_. As a control, THP-1 and Caco-2 cells were plated in the absence of mEVs or with the same amount of 1× PBS. After 48 h, media was removed, replaced with MTS reagent according to manufacturing protocol, and incubated for 2 h at 37 °C with 5% CO_2_. The plates were then analyzed by measuring the absorbance at 490 nm using a microplate reader. As reported in Figure 3, 10^11^ mEVs showed a significant reduction in cell viability in both cell cultures; however, this effect already disappeared on Caco-2 cells at 10^10^ mEVs while persisted on THP-1 cells until 10^9^.

This different effect of 10^10^ and 10^9^ mEVs concentrations on Caco-2 and THP-1 cells viability can be explained by considering the different amounts of plated cells. In fact, after 21 days of differentiation, the Caco-2 cells number is greater than 0.1 × 10^6^ THP-1 cells plated the day before the experiment. Based on this result, to further analyze the effect of mEVs in modulating the inflammation in the co-culture model reported in this work, two different concentrations were used: 10^8^ and 10^10^. The first one had no cytotoxic effect on both cell lines, while 10^10^, although it caused a THP-1 viability reduction, probably induced a better modulation of inflammation in Caco-2 cells. We performed a cell viability assay by quantification of LDH release to evaluate a possible mEV cytotoxic effect on THP-1 and/or Caco-2 cells and, at the same time, demonstrate effective damage on the cell membrane due to the inflammation treatment (C1). The LDH release in Caco-2 or THP-1 monocultures was used as the baseline, while C0 was used as control of a stable co-culture. After the inflammation establishment (C1) and the 24 h mEV treatment (C2a and C2b), 50 µL of cell-free supernatant was removed from every sample from the BL compartment to evaluate the LDH release from THP-1 cells and from the AP compartment to evaluate the Caco-2 damage. In Figure 4, the cytotoxicity percentage is reported, according to datasheet instruction from the CyQUANT LDH kit. In detail, the ratio is expressed as a release of LDH amount normalized to maximum LDH release in the monoculture. The reported data for both cell lines showed a significant increase in the LDH release in the C1 condition compared to C0 and no significant LDH release in Caco-2 monoculture. About THP-1 cells, the cell damage reported could be due to PMA treatment. However, it is worth noticing that there was no significant increase in LDH release, and consequently cell damage, when mEVs were administered to the co-culture, confirming the absence of a cytotoxicity effect of the mEVs even at 10^10^.

### 3.3. Differential Caco-2 Gene Expression Induced by mEVs after Inflammation

As the first result, from the RT-qPCR on cDNA reverse-transcribed from the Caco-2 RNA, the establishment of a pro-inflammatory environment was observed, with a gene expression increase in most of the cytokines tested. *CXCL8*, *IL1B*, *TNFA*, *IL12A*, *IL23A*, and *TGFB1* resulted significantly up-regulated in Caco-2 cells in the inflamed co-culture compared to those of the basal co-culture (Figure 5 and Appendix A). At the same time, a significant upregulation of *NOS2* and *MMP9* was detected, also revealing the induction of oxidative stress and cell damage. Concerning mEV administration, gene expression was evaluated after 24 h from the mEV addition to Caco-2 cells (C2a) and both Caco-2 and THP-1 cells (C2b). Partial restoration of initial co-culture conditions was detected when mEVs were administered at 10^10^ concentration. Indeed, at this concentration, a decrease in most of the tested cytokines was found after mEV administration compared to the inflamed co-culture, although a statistical significance was reached only for *TNFA* (in both C2a and C2b administration typologies) and *TGFB1* when mEVs were given to both Caco-2 and THP-1 (C2b administration). Moreover, a gene expression decrease in *MMP9* and an upregulation of *MUC2* and *TJP1* in both types of administration (C2a and C2b) compared to the inflamed co-culture was shown, indicating a recovery of cellular homeostasis and, therefore, potential beneficial effects on the intestinal mucosa. Concerning the 10^8^ concentration, no statistically significant differences were detected after the mEV administration compared to the inflamed co-culture (Appendix A) except for *MUC2* that increased after mEV administration to Caco-2 cells. These data seem to indicate a dose-dependent effect of mEV administration. This is also confirmed by a statistically significant increase in *IL12A* and *NOS2* exclusively with 10^10^ and solely when both Caco-2 and THP-1 were treated.

### 3.4. Cytokine Production Modulation by mEV Treatment

Finally, multiplex ELISA was employed to evaluate levels of nine key immune cytokines in co-culture supernatants. Supernatants were collected from AP or BL compartments to quantify cytokines released by Caco-2 or THP-1, respectively, in response to different stimuli.

Stimulation with IFN-γ and LPS resulted in increased release of pro-inflammatory cytokines such as IL-17, IL-1β, IL-6, TNF-α by either Caco-2 or THP-1, although with statistical significance only in the latter. For both cell types, higher levels of IL-8 and IL-12 were detected in culture supernatants of C1 compared to C0, although without statistical significance (Figure 6 and Figure 7).

Cytokine levels were assessed after 24 h administration of different mEV concentrations (10^10^ or 10^8^) to culture supernatants: to Caco2 cells only (AP, C2a condition) or to both Caco2 and THP-1 (C2b condition). Before treatment, cells were subjected to inflammatory stimuli (IFN-γ/LPS) as described in material and methods.

Our data revealed that administration of 10^10^ mEVs to C1 culture supernatants determined a marked decrease in the level of several cytokines in Caco-2 culture supernatants: IL-1β, IL-10, IL-12B, IL-12, and IL-17. A reduction was also observed in IL-8 and TNF-α levels, although without statistical significance (Figure 6).

Similar results were observed in THP-1 supernatants: administration of 10^10^ mEVs determined a reduction in several cytokines levels such as IL-8, IL-12B, and IL-12 (Figure 7).

Administration of 10^8^ mEVs to inflamed culture had a weaker impact on both cell types compared to 10^10^ mEVs. As displayed in Appendix A, only IL-1β levels in Caco-2 were decreased by 10^8^ mEV treatment. In addition, we investigated whether mEVs addition to both cell types (C2b condition) had a stronger effect compared to the only Caco2 cells treatment (C2a condition). As displayed in Figure 6 and Appendix A, C2a and C2b presented similar cytokines levels.

## 4. Discussion

Innovative approaches are of pivotal importance for diseases treatment, especially when the pathogenesis is difficult to comprehend and thus resolving therapies are lacking. IBD is the emblem of these pathologies; indeed, various therapeutic approaches, such as aminosalicylates, corticosteroids, immunomodulators, apheresis therapy, calcineurin inhibitors, and cytokines antagonists (i.e., TNF inhibitors) have been attempted, although poor or not entirely satisfactory results have been reached [56]. From this point of view, EVs are promising agents because they can induce various modifications in recipient cells, including the modulation of the immune response and inflammation, key players in the onset of IBD. Furthermore, the variety of molecules contained in the EVs could act on damaged enterocytes following chronic inflammation, improving the mucosal functionality and restoring a situation of homeostasis. In this way, EVs constitute a likely resolutive therapy, acting at different levels through their uptake by all intestinal players. Our research group has recently investigated the potential of EVs derived from milk, where bioactive molecules contained in the cargo such as amino acids, nucleotides, nucleosides, mRNAs, and small RNAs have been fully characterized [47,48].

This work aimed to understand whether these potentials could be translated into concrete actions in improving an inflammatory condition and the enterocyte functionality. To verify this hypothesis, the effect of mEVs was tested on a cellular model of intestinal inflammation. After bovine mEV isolation and characterization, we tried to establish the best concentration to be administered to the cells to have the maximum effect with the minimum degree of toxicity. To do this, a viability assay using MTS reagent was performed, testing from 10^4^ to 10^11^ mEV quantities. As a result, 10^8^ showed no cytotoxic effect on both cell lines, while 10^9^–10^10^ mEVs induced a viability reduction exclusively in THP-1 cells (Figure 3). Despite the decrease in THP-1 cells’ vitality, the higher mEV quantity could increase the probabilities of modulating the inflammation on Caco-2 cells. For this reason, the two chosen concentrations to test mEV effects were 10^8^ and 10^10^.

In Caco-2 co-cultured with THP-1, the LPS and IFN-γ treatment for THP-1 cells and the IFN-γ stimulus for Caco-2 cells were effective in the impairment of Caco-2 cell membrane as reported from LDH assay (Figure 4), and in the establishment of an inflammatory environment, which is observable from the increased transcription of pro-inflammatory cytokine genes compared to the not stimulated co-culture (Figure 5 and Appendix A, C0 vs. C1). Concerning the expression levels of tested genes after administering mEVs to the stimulated co-culture, the 10^10^ concentration (Figure 5, C2a vs. C1, C2b vs. C1) was more effective than 10^8^ (Appendix A, C2a vs. C1, C2b vs. C1). In fact, statistically significant gene expression variations were observed in the 10^10^ concentration only. Fewer particles were effective only on the *MUC2* up-regulation when Caco-2 cells were treated, indicating an improving effect of the higher concentration of mEVs than the lower one. Moreover, mEV treatment inhibited pro-inflammatory cytokines release by Caco-2 cells, reaching a more evident significant result for the 10^10^ concentration (Figure 6) than 10^8^ (Appendix A). Finally, no differences were detected among C2a and C2b types of mEV administration. In THP-1 cells, 10^10^ mEVs (Figure 7) were more effective than 10^8^ (Appendix A) and significantly decreased the amount of IL-12, IL-12B, and IL-17 cytokines after 24 h from the administration.

Concerning epithelial barrier function, mEVs seem to act at different levels, influencing other players. Metalloproteinases have been found to play pivotal roles in the pathogenesis of intestinal inflammations [57]. Metalloproteinases are involved in many processes related to tissue repairs such as angiogenesis and wound healing [58], acting in the physiologic extracellular matrix (ECM) turnover [59] and inducing inflammatory cytokines production. Dysregulation of their expression can cause a prolonged inflammatory response [60]. Among MMPs, MMP-9 seems to be one of the major inflammatory metalloproteinases in IBD patients [61]. Indeed, its inhibition by siRNA treatment or genetic knockdown prevented intestinal inflammation in a murine model of colitis [62], and therapies based on MMP-9 antagonists have been approached, although with controversial results [63]. In the herein used co-culture model of intestinal inflammation induced by LPS and IFN-γ, the up-regulation of *MMP9* after the inflammation stimuli was observed, allowing to test the mEV effect on this pivotal protein. *MMP9* levels in Caco-2 cells decreased after administering mEVs at 10^10^ to Caco-2 and both Caco-2 and THP-1 cells, indicating inhibition of this metalloproteinase by mEV cargo that could be effective in reducing or preventing mucosal damage. Lower mEV concentrations were not able to induce a down-regulation of this gene, inducing to hypothesize a dose-dependent effect of mEVs. Inflammation, in fact, is often associated with increasing TJ intestinal permeability, and MMP-9 can mediate this phenomenon in vitro and in vivo, consequent to the increased expression of the MLCK gene and protein [64], which is linked to the activation of NF-κB [65]. NF-κB is known to induce the expression of IL-8, while MMP-9 is able to proteolytically process *CXCL8* into more potent isoforms [66]; indeed, elevated *CXCL8* expression has been abundantly proven in IBD conditions [67], produced by a variety of cells such as neutrophils, macrophages, fibroblasts, and intestinal epithelial cells, inducing an aberrant leukocyte chemoattraction, correlated with the severity of inflammation [68].

*CXCL8* was up-regulated in Caco-2 cells after inflammation induction, and its protein levels increased in both Caco-2 and THP-1 cells. Interestingly, after the incubation with 10^10^ mEVs, a gene expression down-regulation and a marked decrease in cytokine production in both Caco-2 and THP-1 cells were evident, although significant in THP-1 only. These data suggest a possible indirect effect on the transcription of this gene and its product given by mEV administration. Moreover, the decrease in IL-8 and its transcriptional inhibition is a key step in IBD patients because the release of IL-8 from colon epithelial cells can contribute to the pathological process of gastrointestinal inflammation and malignancies [69]. Additionally, IL-8 is involved in neutrophil attraction through the binding to the CXCR2 receptor [70]. Thus, reducing its production might be a therapeutic strategy in reducing neutrophil-induced inflammation, thus alleviating symptoms. A cytokine whose expression has been found correlated with those of IL-8 is TNF-α; in fact, TNF-α is a pleiotropic pro-inflammatory cytokine implicated in a wide range of cellular processes, thus being an inflammation key mediator and it was seen to be highly expressed in IBD, being produced by both T helper 1 (Th1) and Th2 lymphocytes as well as by macrophages [68,71]. Moreover, TNF-α, as well as IFN-γ, induce barrier dysfunction by modifying transepithelial electrical resistance and TJ permeability through decreased expression of the barrier tightening claudin and occludin [72]. Due to TNF-α’s pivotal role in inflammatory mediation, many therapies aimed to silence its expression have been tried and are a reality at the moment [56,71]. Our experiment showed a significant strong *TNFA* gene expression restoration to basal levels in inflamed Caco-2 cells after treatment with 10^10^ mEVs and a trend of reducing the corresponding protein, suggesting a potentially robust silencing of this crucial cytokine. In addition, in this case, 10^8^ treatment was not able to induce the same *TNFA* gene expression decrease observed for the highest concentration used.

Another important pro-inflammatory cytokine is IL-1β, which has been found to exacerbate IBD in both experimental colitis and colitis in humans when its secretion is increased [73]. Indeed, IL-1β promotes Th17 responses, suggesting synergistic interactions with IL-23 signals that sustain innate and adaptive inflammatory responses in the gut; moreover, the IL-1β regulation of Th17 can be induced by a MyD88-dependent microbial signal [74]. In addition, in this case, particular *IL1B* polymorphisms were significantly associated with the risk of IBD [75,76], conferring to the IL-1β drug silencing a potential therapy role, especially for patients caring these variants or for those with deep ulceration and who do not respond to several different current therapeutics [77]. In this study, an increase in *IL1B* gene expression in Caco-2 cells and IL-1β release for both Caco-2 and THP-1 cells was observed after pro-inflammatory stimuli induction. The 10^10^ mEV treatment was effective in drastically reducing its production in Caco-2, while no significant differences were observed in THP-1. Moreover, IL-1β plays a role in intestinal permeability through the activation of miR-200c-3p machinery that increases the degradation of the occluding mRNA, thus increasing TJ permeability [78]. This evidence indicates that the IL-1β reduction mediated by mEV treatment might be promising also for intestinal homeostasis improvement or damage prevention.

Finally, another key regulator in IBD is IL-10, an anti-inflammatory cytokine produced by monocytes, B cells, T cells, as well as some other cells. IL-10 induces the immune response of Th2 cells, but in the gut, it is mainly produced by macrophages, B cells, and colonic lamina propria cells following enteric bacteria stimulation [79]. Probably, the absence of these cells in our model, as well as the interaction with the microbiome, explains the lack of IL-10 increase after mEV administration in Caco-2. On the other hand, THP-1 cells showed a growing trend of this cytokine, even if it did not reach significance. Indeed, it has previously been established that the role of IL-10 in IBD depends on IL-10 signaling in macrophages [80].

Regarding the TJ permeability consequent to the chronic inflammation and its re-establishment to physiological levels, our results showed an important increase in TJP1 gene expression after 10^10^ mEV administration to Caco-2 and Caco-2/THP-1 cells, which might take part in normal mucosal permeability restoring. In this context, we also evaluated MUC2 and TFF3 modulation after mEV treatment. MUC2, in fact, is essential in preventing pathogen-induced epithelial injury, constituting the main component of the viscous mucus layer on the surface of the gastrointestinal tract that mediates innate immunity protective functions [81]. A thickness reduction in the colonic mucus was observed in IBD patients [82]. A MUC2 deficiency was shown in colonic bacteria in contact with epithelium, resulting in an increased level of inflammatory cytokines [81]. MUC2 seems essential for protecting the mucosal barrier and conferring the proper functioning, especially in maintaining the homeostatic cross-talk between microbiota and host intestinal cells. Interestingly, a strong statistically significant *MUC2* up-regulation was observed in our experiment in both C2a and C2b mEV administrations for 10^10^ concentration, but at variance from the other results, a statistical significance of *MUC2* increase expression was also detected for the 10^8^ administration to Caco-2 cells. Together with MUC2, TFF3 is principally expressed in intestinal goblet cells where it participates in mucosal regeneration after injury, in addition to mucosal protection of healthy tissue and it has been proposed its ability to inhibit inflammatory cytokines [83]. Our results showed no differences in *TFF3* expression in inflamed cells compared to basal co-culture or mEV treatment, probably because prolonged inflammation is needed to modulate its gene expression or the real injuries that are established during chronic inflammations are scarcely represented in this cellular model. These results indicate a partial restoration of initial co-culture conditions, especially when mEVs were administered at 10^10^ concentration.

In our experiment, an increase in *TGFB1* expression was observed after inflammatory stimuli, and a statistically significant down-regulation was detected after 10^10^ mEV administration to both Caco-2 and THP-1 cells, suggesting a possible regulatory role of mEV cargo in balancing this cytokine secretion, favoring the maintenance of the proper balance. In fact, TGF-β1-signaling is strongly implicated in gut homeostasis and regulates many mucosal cell types. For example, TGF-β1 decreases dendritic cells maturation and antigen presentation, increases monocyte recruitment, and reduces macrophages responsiveness [84]. Concerning lymphocytes, TGF-β1 increases B cell reaction and promotes IgA secretion. Furthermore, in T cells, low concentrations of TGF-β1 induce Treg differentiation via the down-regulation of the IL-23 receptor, contributing to the immune tolerance induction. Conversely, the up-regulation of IL-23 is associated with high TGF-β1 levels, promoting Th17 polarization [84]. In this context, the observed modulation of *TGFB1* induced by mEVs might be important for pushing this balance in favor of an anti-inflammatory action by this cytokine. TGF-β1-signaling disruption is associated with the development of intestinal inflammation. However, TGF-β1 levels are increased in IBD tissues as a response to the ongoing inflammation [85]. Moreover, TGF-β1 stimulates stromal cells to produce fibrogenic mediators and regulators of extracellular matrix deposition, promoting the differentiation of mesenchymal cells in myofibroblasts, thus being considered as a major fibrogenic cytokine. Indeed, intestinal fibrosis and strictures have been associated with an uncontrolled TGF-β1 activity due to its increased expression that determinates an excessive accumulation of extracellular matrix proteins [86]. Thus, the *TGFB1* modulation by mEVs might prevent excessive fibrosis after injuries consequent to inflammation. The mEV regulatory function on *TGFB1* expression and other previously mentioned genes seems to be dose-depended and seems to be particularly related to vesicular cargo action on macrophages, since only when administered to both Caco-2 and THP-1 cells there is a significant change.

As expected, a strong *NOS2* up-regulation in inflamed C1 Caco-2 cells compared to C0 was detected. Surprisingly, the 10^10^ mEV treatment was not able to reduce *NOS2* expression, which further increased when both cell types were treated (C2b). This modulation, together with the fact that 10^8^ treatment (Appendix A) did not induce an ulterior *NOS2* up-regulation compared to the C1 condition, suggests a dose-dependent effect. This could be explained by the fact that mEVs are rich in arginine [47], a substrate for NOS2 from which these enzymes synthetizes NO, conferring strong anti-microbial and immune-regulatory activities [87]. Arginine seems to be able to accelerate resolution from colitis [88]; however, the co-culture model used in this study is not able to evaluate its influence on intestinal microbiota diversity restoration, which seems to be one of the major mechanisms of action [88,89]. On the other hand, an abundant arginine availability could induce an up-regulation of its catalytic enzyme NOS2, explaining what was observed in Caco-2 cells [90]. Finally, the M1 macrophage polarization induced by the use of LPS and IFN-γ correlates with the strong increase in *IL12A*. IL-12 is a member of the IL-12 family together with IL-23, IL-27, and IL-35. These are dimeric cytokines consisting of two proteins encoded by different genes. In particular, IL-12 is composed of IL-12A and IL-12B subunits, while the association of IL-12A and EIB3 constitute IL-35. Therefore, the observed *IL12A* gene expression increase could be related to IL-12 and/or IL-35 protein production; however, IL-35 is preferentially secreted by Treg cells [91]. The up-regulation of *IL12A* by the 10^10^ mEV administration to Caco-2 and THP-1 cells could be considered a sign of equilibrium restoration. *IL12A* up-regulation, indeed, was shown in patients affected by IBD during the remission phase compared to the flare-up [92]. The role of IL-12A is not entirely clear in IBD onset, although it seems to orchestrate the initial, primarily innate immune cell-driven inflammatory reaction triggered by exposure to bacteria in response to intestinal barrier damage [93]. Moreover, IL-12A has been found to regulate inflammation since its deletion aggravates LPS-induced cardiac injury and cardiac dysfunction by exacerbating the imbalance of M1 and M2 macrophages [94]. Concerning our ELISA results, the parallel decrease in IL-12B subunit and IL-12 cytokine release in both Caco-2 and THP-1 supernatants not only demonstrates that mEV treatment drastically reduces active IL-12 but strengthens our hypothesis related to the increased production of IL-35. A recent study assessed the therapeutic potential of IL-35 recombinant protein in a mouse model of colitis; the IL-35 treatment decreased the infiltration of macrophages, CD4+ T and CD8+ T cells and promoted the Treg cell infiltration, increased the IL-10 production and decreased IL-6, TNF-α, and IL-17A, alleviating the inflammation in acute colitis [95]. Concerning active IL-12 release, it is known that this cytokine plays a key role in the differentiation of naive T cells to Th1 cells, and a high proportion of IL-12 genetic risk loci for IBD have been found [96]. Other than being an IL-12 subunit, IL-12B is part of IL-23, another crucial regulatory cytokine in chronic intestinal inflammation [97]. These characteristics have induced many approaches to IBD therapy to inhibit both the IL-12 and IL-23 signaling pathways, such as the use of monoclonal antibodies targeting p40 [98] or through epithelial HIF-stabilization that regulate T helper cell populations via IL-12 cytokine family signaling [99]. The treatment with mEVs can modulate these pivotal pathways, inducing ameliorating effects since the 10^10^ mEV concentration was effective on both Caco-2 and THP-1 cells to induce a reduction in active IL-12 and IL-12B cytokines. Moreover, IL-23 is especially important for maintenance and expansion of the pro-inflammatory Th17 lineage via the upregulation of IL-17, RORγt, TNF, IL-1, and IL-6, indicating the important role of the IL23/IL17 axis IBD [100]. Our results on IL-17 dosage in Caco-2 supernatant showed a tremendous decrease in this cytokine production through the administration of 10^10^ mEVs. This is an important finding since IL-17 has been recently highlighted as a pro-inflammatory cytokine implicated in the onset of many autoimmune diseases, including CD. Immune therapy approaches using anti-IL-17 antibodies have been applied. They induce beneficial outcomes but also provoke important side effects [101].

## 5. Conclusions

This study aimed to evaluate the anti-inflammatory and immunomodulating effects of mEVs derived from bovine milk on an in vitro co-culture model of intestinal inflammation. *CXCL8*, *IL1B*, *TNFA*, *IL12A*, *IL23A,* and *TGFB1* genes together with *NOS2* and *MMP9* were significantly up-regulated in Caco-2 cells in the inflamed co-culture; these data were supported by the determination of cytokine levels in culture supernatants, where IL-17, IL-1β, IL-6, TNF-α increased levels were observed. The 10^10^ mEV treatment induced a partial restoration of initial co-culture conditions, highlighted by a decrease in most of the tested cytokines, both for gene expression and protein production, known to play a pivotal role in IBD. Moreover, a gene expression down-regulation of *MMP9* and an up-regulation of *MUC2* and *TJP1* indicate an attempt to restore cellular homeostasis and mucosal functions. In conclusion, the in vitro tests of mEVs efficacy are promising for the use of mEVs as adjuvant therapy for the IBD syndrome, even if they will have to be tested in more detail, especially in disease models that consider the interaction with the intestinal microbiome and possible in vivo tests.

## Figures and Tables

**Figure 1 biomedicines-10-00570-f001:**
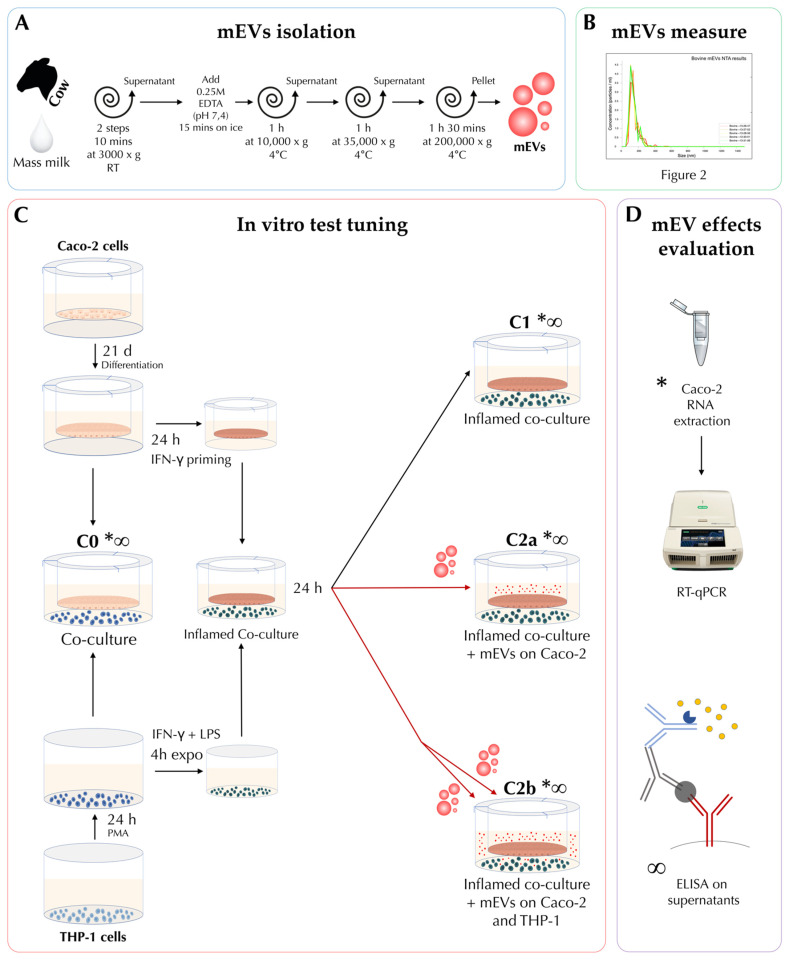
Experimental design and main passages of methods at a glance. The (**A**) box shows milk-derived extracellular vesicle (mEV) isolation steps from bovine milk; in the (**B**) box, there is a reduced representation of the nanoparticle tracking analysis (NTA) results for concentration and size assessment of mEVs; the (**C**) box introduces the main passages for co-culture preparation and experimental conditions: C0 (co-culture); C1 (co-culture after 48 h of inflammation); C2a (24 h inflamed co-culture after 24 h of mEV inoculation in the apical (AP) side); and C2b (24 h inflamed co-culture after 24 h of mEV inoculation in both in the AP and BL side). Box (**D**) shows the types of test that were performed on Caco-2 cells and supernatants: * indicates the conditions where gene expression of Caco-2 cells was tested through RNA extraction and RT-qPCR evaluation; while the Caco-2 and THP-1 supernatants, tested through ELISA, are marked with the ∞ PMA: phorbol 12-myristate 13-acetate; IFN-γ, interferon gamma; LPS, lipopolysaccharides.

**Figure 2 biomedicines-10-00570-f002:**
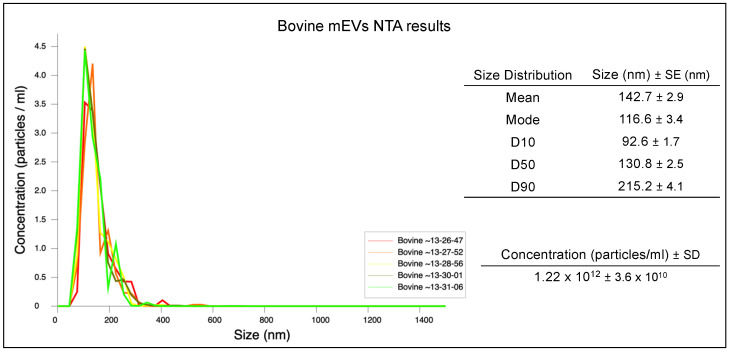
NTA result of mEVs isolated from bovine milk. The peak of the mEV size distribution was 105 nm.

**Figure 3 biomedicines-10-00570-f003:**
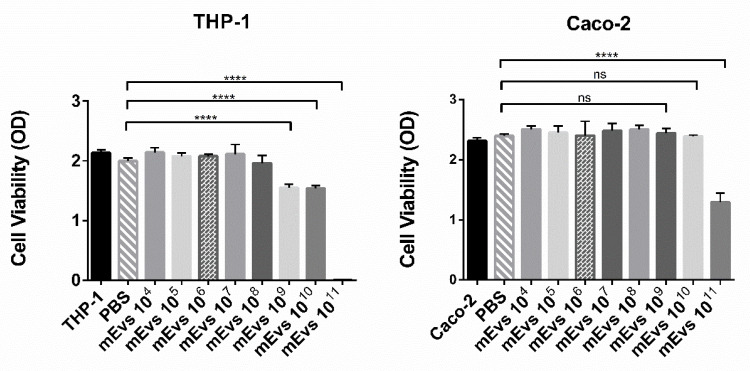
The 3-(4,5-dimethylthiazol-2-yl)-5-(3-carboxymethoxyphenyl)-2-(4-sulfophenyl)-2H-tetrazolium, inner salt (MTS) cell viability assay. Twenty-one (21) days differentiated Caco-2 cells and THP-1 cells were seeded in a 96-well plate in the absence and presence of different amounts of mEVs from 1 × 10^4^ to 1 × 10^11^. Cells were incubated for 48 h at 37 °C under 5% CO_2_. After 2 h incubation with MTS reagent, absorbance at 490 nm was measured using a microplate reader. The data are representative of 3 independent experiments. **** *p* < 0.0001, ns *p* > 0.05.

**Figure 4 biomedicines-10-00570-f004:**
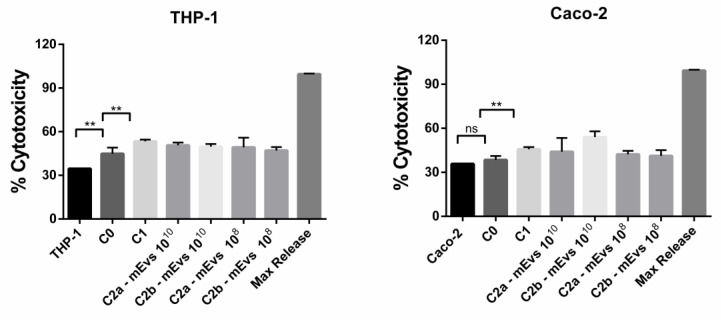
The lactate dehydrogenase (LDH) assay in the presence of mEVs. LDH release from Caco-2 or THP-1 in the presence or absence of two different concentrations of mEVs. LDH was measured by optical density measurement at 490 nm in the supernatant recovered after 24 h mEV treatment. Caco-2 and THP-1 alone were used as a positive control, C0 co-culture before inflammation, C1 co-culture after 24 h of inflammation, C2a mEV administration to Caco-2 cells in the AP compartment of 24 h inflamed co-culture, C2b mEV administration to Caco-2 and THP-1 cells in the AP compartment and BL compartment respectively of mEVs after 24 h inflamed co-culture. The figure reports data from the triplicate experiment analyzed by a one-way ANOVA test, with mean ± SD values. ** *p* < 0.01 and ns *p* > 0.05.

**Figure 5 biomedicines-10-00570-f005:**
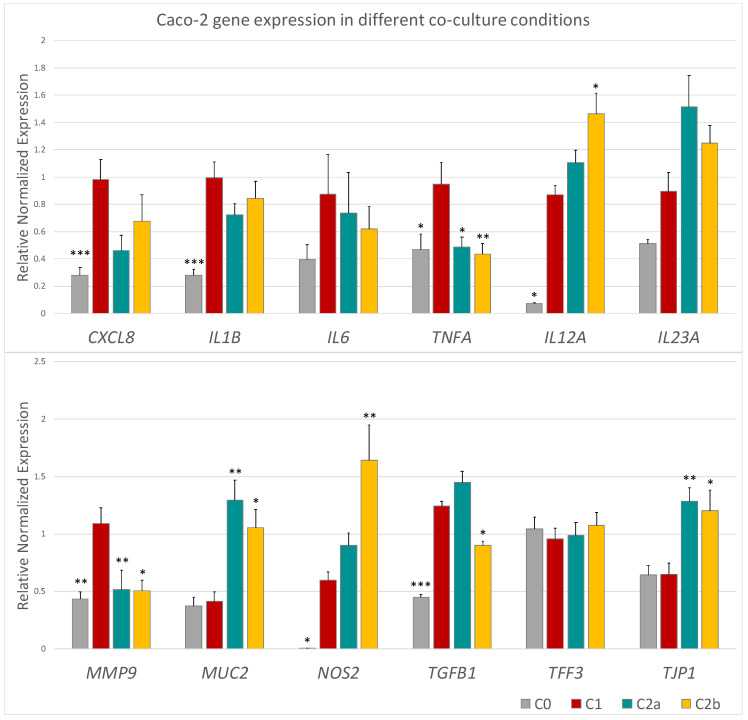
Histogram of Caco-2 tested gene expression in the different culture conditions:co-culture of Caco-2 and THP-1 cells in basal conditions (C0—gray), inflamed co-culture (C1—red), 10^10^ mEV administration to Caco-2 cells in inflamed co-culture (C2a—green), and 10^10^ mEV administration to Caco-2 and THP-1 cells in inflamed co-culture (C2b—yellow). Differences (others vs C1) were evaluated using the Kruskal–Wallis test and applying the post-doc Dunn’s multiple comparison test. The asterisks indicate the statistical significance: * *p* < 0.05, ** *p* < 0.01 and *** *p* < 0.001.

**Figure 6 biomedicines-10-00570-f006:**
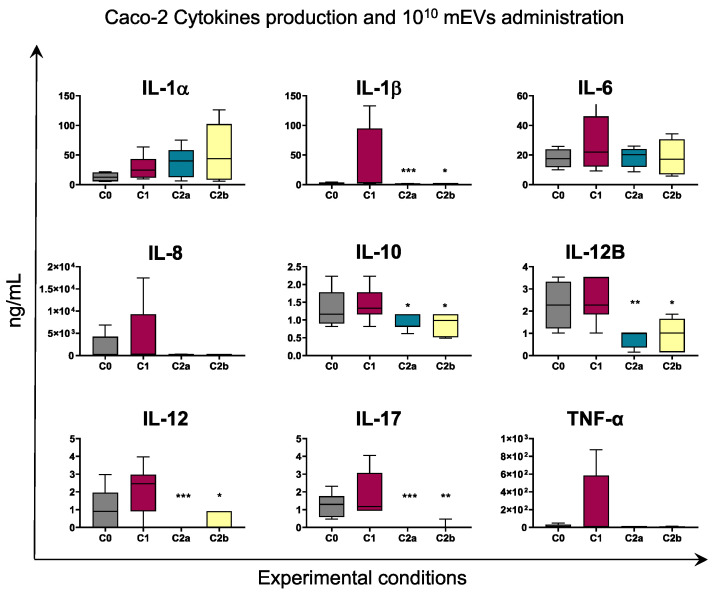
The 10^10^ mEV impact on cytokine production by Caco-2. Caco-2 and THP-1 cells were left untreated (basal condition, C0) or stimulated with IFN-γ and LPS (inflamed co-culture, C1). At 24 h post stimulation, 10^10^ mEV suspension was added to Caco-2 cells in inflamed co-culture (C2a) or to both Caco-2 and THP-1 cells in inflamed co-culture (C2b). A total of 24 h later, culture supernatants were collected, and levels of cytokines were determined through ELISA. Data are presented as box-and-whisker plots displaying median and interquartile range (boxes) and minimum and maximum values (whiskers). Values of C0, C2a, C2b were compared to C1, using an ANOVA followed by Dunnett’s multiple comparison test or a Kruskal–Wallis test followed by Dunn’s multiple comparison test; * *p* < 0.05, ** *p* < 0.01 and *** *p* < 0.001.

**Figure 7 biomedicines-10-00570-f007:**
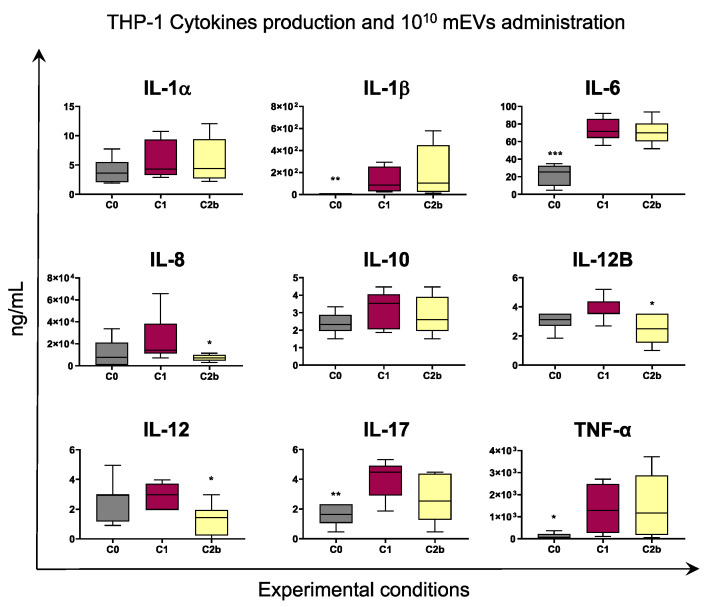
The 10^10^ mEV impact on cytokine production by THP-1. Co-culture of Caco-2 and THP-1 cells were left untreated (basal condition, C0) or stimulated with IFN-γ and LPS (inflamed co-culture, C1). At 24 h post stimulation, 10^10^ mEV suspension was added to both Caco-2 and THP-1 cells in inflamed co-culture (C2b). A total of 24 h later, culture supernatants were collected, and levels of cytokines were determined through ELISA. Data are presented as box-and-whisker plots displaying median and interquartile range (boxes) and minimum and maximum values (whiskers). Values of C0, C2a, C2b were compared to C1, using an ANOVA followed by Dunnett’s multiple comparison test or a Kruskal–Wallis test followed by Dunn’s multiple comparison test; * *p* < 0.05, ** *p* < 0.01 and *** *p* < 0.001.

**Table 1 biomedicines-10-00570-t001:** Primer set sequences for target genes and reference.

	Gene	Primer Sequences	Amplicon Length	Accession Number
Target genes	*CXCL8*	For-5′-CTCTCTTGGCAGCCTTCCT-3′ Rev-5′-TTGGGGTGGAAAGGTTTGGA-3′	118	NM_000584.4
	*IL1B*	For-5′-CAGGGACAGGATATGGAGCA-3′ Rev-5′-ACGCAGGACAGGTACAGATT-3′	122	XM_017003988.2
	*IL6*	For-5′-GAGAGTAGTGAGGAACAAGCC-3′ Rev-5′-GGTCAGGGGTGGTTATTGCA-3′	106	NM_000600.5
	*TNFA*	For-5′-CCTCAGCCTCTTCTCCTTCC-3′ Rev-5′-GGCTTGTCACTCGGGGTT-3′	122	NM_000594.4
	*IL12A*	For-5′-ACCACTCCCAAAACCTGCT-3′ Rev-5′-CCAATGGTAAACAGGCCTCC-3′	150	NM_000882.4
	*IL23A*	For-5′-GAAGCTCTGCACACTGGC-3′ Rev-5′-TGTTGTCCCTGAGTCCTTGG-3′	143	NM_016584.3
	*MMP9*	For-5′-AAGGCGCAGATGGTGGAT-3′ Rev-5′-TCAACTCACTCCGGGAACTC-3′	150	NM_004994.3
	*MUC2*	For-5′-CACCTGGCTGTGCTTAACG-3′ Rev-5′-GCGGGAGTAGACTTTGGTGT-3′	99	NM_002457.4
	*NOS2*	For-5′-ACCTCAAGCTATCGAATTTGTCA-3′ Rev-5′-CTCATCTCCCGTCAGTTGGT-3′	136	NM_000625.4
	*TGFB1*	For-5′-GTTGTGCGGCAGTGGTTG-3′ Rev-5′-AGTGTGTTATCCCTGCTGTCA-3′	89	NM_000660.7
	*TFF3*	For-5′-GAGTCCTGAGCTGCGTCC-3′ Rev-5′-GCACGGCACACTGGTTTG-3′	138	NM_003226.4
	*TJP1*	For-5′-GCAACTAAGGAAAAGTGGGAAAA-3′ Rev-5′-GGTTCAGGATCAGGACGACTTA-3′	90	NM_001301025.3
Reference genes	*RPLP0*	For-5′-CAGGGAAGACAGGGCGAC-3′ Rev-5′-CCACATTGTCTGCTCCCACA-3′	101	NM_001002.4
	*RPL37A*	For-5′-ATGAAGAGACGAGCTGTGGG-3′ Rev-5′-GTCTTCTGATGGCGGACTTT-3′	118	NM_000998.5
	*RPS14*	For-5′-ATCACCGCCCTACACATCAA-3′ Rev-5′-ATCCGCCCGATCTTCATACC-3′	119	NM_005617.4

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
