# Peer review of "Cow Milk Extracellular Vesicle Effects on an In Vitro Model of Intestinal Inflammation"

_biomedicines, 2022, doi:10.3390/biomedicines10030570_

Round 1

Reviewer 1 Report

I think the paper is well organized and the methods emplyed are well described. However I have some suggestions for the authors:

1- Figure 1: some of the captions are partially covered by the sketches and it is impossible to read them; moreover there is a typing mistake in the word "interferon-γ"

2- On page 6, lines 237-238: the sentence "For testing co-coltures was performed LDH assay" should be replaced with "For testing co-cultures an LDH assay was performed".

3- On page 6, lines 245-246: The sentence "As control, were used non treated monoculture cells...." should be replaced woth "As control, non treated monocoltured cells were used..."

4- On page 6, lines 247-248: The sentence "After incubation were added 50 μL of STOP solution to the plate...." should be replaced with "After incubation, 50 μL of STOP solution were added to the plate..."

5- On page 9, line 366: the verb "shawn" should be replaced wih the simple past "showed"

6- On page 12, line 460: "revelated" should be replaced with "revealed"

7- Finally, I think that in the introduction it should be mentioned the possibility to use EVs as drug carriers; among the literature I suggest the authors to cite the following papers:

  • Lu, M.; Huang, Y.Y. Bioinspired exosome-like therapeutics and delivery nanoplatforms. Biomaterials 2020, 242, 119925.
  • Ailuno, G.; Baldassari, S.; Lai, F.; Florio, T.; Caviglioli, G. Exosomes and Extracellular vesicles as emerging theranostic platforms in cancer research. Cells 2020, 9, 2569

Reviewer 2 Report

In this manuscript entitled “Cow Milk Extracellular Vesicle effects on an in vitro model of 2

intestinal inflammation” by Mecocci et al., the authors have explored the anti-inflammatory effect of mEVs using an in vitro cell culture model. The authors tested for toxicity of mEVs and their ability to alter genes expression at the mRNA level using quantitative PCR and protein levels using ELISA. Observed results suggest that mEVs rescue the co-culture system from inflammation generated by LPS and IFN- γ. This is an exciting observation considering the increased prevalence of inflammation associated with IBD. Though the in vitro observation for mEVs is promising, this warrants a thorough in vivo investigation as the microenvironment in the gut plays a significant role in orchestration the inflammatory response. However, the current study in the in vitro perspective is exciting and will gain attention. The study is well designed and executed; however, I request the authors correct the writing part as there are several grammatical errors and typos throughout the manuscript, making it very hard to understand the significant results observed in this study.  I suggest the authors revise the manuscript thoroughly and submit it for further revision.

Below are some of my specific comments, suggestions, and questions

  1. Line 34. Remove “resulted” and include “were”

  1. Simplify the abstract; many sentences in the abstract are stretched without any need.

  1. Line 88 smoking? What medical conditions?

  1. Line 101: the ability of EV to regulate

  1. Extensive English corrections are required for the introduction.

  1. The schematic figure for methods is not helpful; please remove it.

  1. Remove line 155-157

  1. Line 175: Before measuring

  1. Line 203: impair

  1. The materials and methods section is well written.

  1. The title of the results section should be modified. The current title looks like the methods section title. The results title should summarize the result in a short line.

  1. What makes EVs toxic to cells?

  1. Does mEV vary from one cow species to other?

  1. In Figure 4, EVs did not protect against inflammation in THP-1 cells and Caco2 cells. How do the authors consider this?

  1. In Figure 5, I do not see a significant difference in TGFB1 in C2a Caco2 cells. Will this affect the inflammatory status of co-culture?

  1. In figure 6, IL-8, IL-17, and IL-12 levels seem to be well below the basal level of C0 cells; what is the author’s justification for this? Further, how significant is the increase of the tested cytokines in C1 cells? I do not see a significant increase in IL-8, IL-10 IL 12B, IL -12, and IL-17. Please explain.  

  1. Can drinking milk provide the effect of EVs in IBD patients? How the current study can be translated into disease treatment in humans.
